# Lymph node volume predicts survival in esophageal squamous cell carcinoma treated with neoadjuvant chemoradiotherapy and surgery

Tzu-Hui Pao[1], Ying-Yuan Chen[2], Wei-Lun Chang[3], Shang-Yin Wu[4], Wu-Wei Lai[2], Yau-Lin Tseng[2], Ta-Jung Chung[5], Forn-Chia Lin[1] *

1 Division of Radiation Oncology, Department of Oncology, National Cheng Kung University Hospital, College of Medicine, National Cheng Kung University, Tainan, Taiwan, 2 Department of Surgery, National Cheng Kung University Hospital, College of Medicine, National Cheng Kung University, Tainan, Taiwan, 3 Department of Internal Medicine, National Cheng Kung University Hospital, College of Medicine, National Cheng Kung University, Tainan, Taiwan, 4 Department of Oncology, National Cheng Kung University Hospital, College of Medicine, National Cheng Kung University, Tainan, Taiwan, 5 Department of Medical Imaging, National Cheng Kung University Hospital, College of Medicine, National Cheng Kung University, Tainan, Taiwan

* fornchia@mail.ncku.edu.tw

**Data Availability Statement:** All relevant data are within the manuscript and its Supporting Information files.

## Abstract

Large primary tumor volume has been identified as a poor prognostic factor of esophageal squamous cell carcinoma (ESCC) treated with definitive concurrent chemoradiotherapy (CCRT). However, when neoadjuvant CCRT and surgery are adopted, the prognostic impact of primary tumor and lymph node (LN) volume on clinical outcomes in ESCC remains to be elucidated. This study included 107 patients who received neoadjuvant CCRT and surgery for ESCC. The volume of the primary tumor and LN was measured using radiotherapy planning computed tomography scans, and was correlated with overall survival (OS), disease-free survival (DFS), and cancer failure pattern. The median OS was 24.2 months (IQR, 11.1–93.9) after a median follow-up of 18.4 months (IQR, 8.1–40.7). The patients with a baseline LN volume > 7.7 ml had a significantly worse median OS compared to those with smaller LN volume (18.8 vs. 46.9 months, p = 0.049), as did those with tumor regression grade (TRG) 3–5 after CCRT (13.9 vs. 86.7 months, p < 0.001). However, there was no association between OS and esophageal tumor volume (p = 0.363). Multivariate analysis indicated that large LN volume (HR 1.753, 95% CI 1.015–3.029, p = 0.044) and high TRG (HR 3.276, 95% CI 1.556–6.898, p = 0.002) were negative prognostic factors for OS. Furthermore, large LN volume was linked to increased locoregional failure (p = 0.033) and decreased DFS (p = 0.041). In conclusion, this study demonstrated that large LN volume is correlated with poor OS, DFS, and locoregional control in ESCC treated with neoadjuvant CCRT and esophagectomy.

**Funding:** This work was supported by National Cheng Kung University Hospital of Taiwan [NCKUH-11203005 to THP and NCKUH-11203051 and NCKUH-11302018 to FCL]. The funder had no role in study design, data collection and analysis, decision to publish, or preparation of the manuscript.

**Competing interests:** The authors have declared that no competing interests exist.

## Introduction

Esophageal cancer is the seventh most common cancer and the sixth leading cause of cancer-related deaths globally [1]. For patients with resectable non-early esophageal cancer, neoadjuvant chemoradiotherapy followed by surgery has been suggested [2]. However, even with tri-modality therapy, more than 40% of patients died from relapsed esophageal cancer [3]. Therefore, identifying prognostic factors and relapse patterns could pave the way for improved outcomes.

Large primary tumor volume has been identified as a poor prognostic factor in esophageal squamous cell carcinoma (ESCC) treated with definitive concurrent chemoradiotherapy (CCRT) [4, 5]. In this setting, large primary tumor volume correlated with inferior local control and overall survival (OS) [4]. However, when neoadjuvant CCRT and surgery are adopted, the prognostic impact of primary tumor and lymph node (LN) volume on clinical outcomes in ESCC remains to be elucidated.

To address this question, we conducted a study of a single-institution cohort of ESCC patients who received neoadjuvant CCRT followed by surgery. The volume of the primary tumor and LN was respectively measured and correlated with the clinical outcomes.

## Materials and methods

### Patients and study design

Electronic medical records of patients with ESCC who were treated with neoadjuvant CCRT and surgery at our institution between 2008 and 2021 were retrieved during October, 2022 to December, 2022. Patients were recruited based on the following criteria: newly pathologically confirmed ESCC without distant metastasis, no prior thoracic radiotherapy, neoadjuvant CCRT using intensity-modulated radiotherapy (IMRT), a radiation dose of 52 Gy or less, and an interval of no more than 3 months between CCRT and surgery. The pre-treatment evaluation for esophageal cancer included esophagogastroduodenoscopy, endoscopic ultrasonography (EUS), computed tomography (CT) of the chest and abdomen, and bone scan. Positron emission tomography-computed tomography (PET-CT) was performed in cases with indeterminate results on CT or bone scan. Cancer staging was based on the seventh edition of the American Joint Committee on Cancer staging system. The data source for this review and the access to participants' information was approved by the Institutional Review Board (IRB) of the National Cheng Kung University Hospital, and it waived the requirement for written informed consent of this study (Approval number NCKUH IRB No.: A-ER-111-327).

### Gross tumor volume delineation and volume measurement

The simulation CT scan for radiotherapy was acquired with a slice thickness of 5 mm and imported into the Eclipse treatment planning system (Varian Medical Systems). The esophagus tumor and involved lymph nodes were delineated on each slice of the planning CT scan with reference to the CT scan, PET-CT, endoscopy and EUS. The gross tumor volume (GTV) of primary tumor (GTVp) and LN (GTVn) was measured using the volume computation function of the Varian Eclipse system (S1 Fig).

### Trimodality therapy

Patients were treated with CCRT using the IMRT technique as previously described [6]. In brief, clinical target volume (CTV) 1, which included GTVp with a 5-cm craniocaudal and 1-cm radial margin along the esophagus, and GTVn with a 1-cm margin, received a dose of 36 Gy. Starting from 2017, CTV 2 was additionally created, consisting of GTVp with a 2-cm

craniocaudal and 1-cm radial margin along the esophagus, and GTVn with a 1-cm margin. CTV 1 and 2 were irradiated sequentially to 36 and 40–50.4 Gy, respectively. The planning target volume was generated by expanding 1 cm around CTV in all directions. Additionally, patients received chemotherapy, nutrition, and supportive care during radiation. Four weeks after CCRT completion, restaging was performed using EUS of the esophagus and CT of the chest and abdomen. Patients with resectable ESCC underwent esophagectomy and lymph node dissection within 3 months of completing CCRT.

## Outcomes

Institutional pathology reports were reviewed for pathologic outcomes which were determined at the time of surgical resection. Tumor regression was evaluated using the Mandard grading system, which categorizes tumor regression into five tumor regression grade (TRG), with TRG 1 indicating fibrosis without residual cancer, whereas TRG 5 indicating the absence of regressive changes [7]. Follow-up evaluations after esophagectomy included clinical examinations and CT scans of the chest and abdomen every 3–6 months for 2 years and every 6–12 months thereafter. Esophagogastroduodenoscopy and EUS were performed annually, while PET-CT or other examinations were done as clinically indicated. OS was measured from the date of surgery to the date of death. Disease-free survival (DFS) was defined as the time from surgery until the first occurrence of cancer relapse or death.

## Statistical analysis

Survival and cancer relapse were estimated using the Kaplan-Meier method and compared between groups by the log-rank test. The factors associated with OS were examined through univariate analysis. Multivariate Cox proportional hazards regression analyses were conducted to determine the independent risk factors of OS, taking into consideration the variables with a trend ($p < 0.3$) in univariate analysis. $P < 0.05$ was considered statistically significant. All statistical analyses were performed using SPSS version 22.0 software and R version 3.5.1 for Windows.

# Results

## Baseline patient characteristics

Of the 131 patients reviewed, 107 patients met the recruitment criteria, while 24 patients were excluded for reasons as follows: histology other than squamous cell carcinoma (n = 6), radiation dose > 52 Gy (n = 6), the interval between neoadjuvant CCRT and surgery being over 3 months (n = 8), and the use of three-dimensional conformal radiation therapy technique (n = 4). Table 1 summarized the demographic and clinical characteristics of the included patients, comprising seven (6.5%) women and 100 (93.5%) men. Ninety-four (87.9%) patients were diagnosed with stage III ESCC. The median volume of esophageal tumor and LN were 42.2 (IQR, 24.7–71.9) and 7.7 (IQR, 3.2–12.9) ml, respectively.

## Treatment

The median radiation dose was 41.4 Gy (range, 36.0–52.0). Fluoropyrimidine-based chemotherapy regimens were used in 100 (93.5%) patients. The majority of patients received either cisplatin (20 mg/m$^2$ daily, on days 1–4) plus fluorouracil (800 mg/m$^2$ daily, on days 1–4) administered intravenously every 4 weeks or cisplatin (25 mg/m$^2$) plus fluorouracil (1000 mg/m$^2$) given intravenously every week. All patients underwent esophagectomy. The median interval between neoadjuvant CCRT and surgery was 45 days (IQR, 37–56).

**Table 1. Demographic and clinical characteristics of patients at baseline.**

| Characteristic | No. of patients (%) |
|---|---|
| **Age (years)** | |
| Median (Range) | 54 (36–79) |
| $\leq$ 54: > 54 | 54 (50.5): 53 (49.5) |
| **Gender** | |
| Male: Female | 100 (93.5): 7 (6.5) |
| **ECOG PS** | |
| 0: 1: 2 | 27 (25.2): 65 (60.7): 15 (14.0) |
| **cStage** | |
| I: II: III | 1 (0.9): 12 (11.2): 94 (87.9) |
| **cT category** | |
| 1: 2: 3: 4 | 4 (3.7): 15 (14.0): 86 (80.4): 2 (1.9) |
| **cN category** | |
| 0: 1: 2: 3 | 5 (4.7): 25 (23.4): 45 (42.1): 32 (29.9) |
| **Tumor location** | |
| U: M: L | 18 (16.8): 40 (37.4): 36 (33.6) |
| U + M (from U to M) | 2 (1.9) |
| M + L (from M to L) | 11 (10.3) |
| **Tumor length (cm)** | |
| Median (Range) | 5.0 (1.0–16.0) |
| $\leq$ 5.0: > 5.0 | 58 (54.2): 49 (45.8) |
| **Esophageal tumor volume (ml)** | |
| Median (Range) | 42.2 (5.8–177.2) |
| $\leq$ 42.2: > 42.2 | 54 (50.5): 53 (49.5) |
| **Lymph node volume (ml)** | |
| Median (Range) | 7.7 (0.0–45.1) |
| $\leq$ 7.7: > 7.7 | 54 (50.5): 53 (49.5) |
| **PET-CT** | |
| Yes: No | 45 (42.1): 62 (57.9) |
| **Smoking** | |
| Yes: No | 94 (87.9): 13 (12.1) |
| **Alcohol** | |
| Yes: No | 95 (88.8): 12 (11.2) |
| **Radiation dose (Gy)** | |
| Median (Range) | 41.4 (36.0–54.0) |
| $\leq$ 41.4: > 41.4 | 84 (78.5): 23 (21.5) |
| **Chemotherapy regimen** | |
| Fluropyrimidine-based | 100 (93.5) |
| Taxane-based | 7 (6.5) |

ECOG PS, Eastern Cooperative Oncology Group performance status; U, upper thoracic and cervical esophagus; M, middle thoracic esophagus; L, lower thoracic esophagus.

## Pathological outcomes after CCRT

Pathological complete response (pCR) was observed in 33 (30.8%) and TRG 1–2 in 57 (53.2%) patients (Table 2). Negative surgical margin was confirmed in 89 (83.2%) patients while the margin was close (< 1 mm) and involved in 15 (14.0%) and 3 (2.8%) cases, respectively. Post-therapy pathological stage was 0 in 33 (30.8%), I in 8 (7.5%), II in 38 (35.5%), and III in 28 (26.2%) cases. Tumor down-staging was observed in 71 (66.4%) patients.

**Table 2. Treatment outcomes and cancer progression pattern after neoadjuvant concurrent chemoradiotherapy and surgery.**

| Variables | No. of patients (%) |
|---|---|
| **Pathological complete response** | |
| Yes: No | 33 (30.8): 74 (69.2) |
| **Tumor regression grade** | |
| 1: 2: 3: 4: 5: Unknown | 36 (33.6): 21 (19.6): 20 (18.7): 17 (15.9): 1 (0.9): 12 (11.2) |
| **Surgical margin** | |
| Negative: Close: Involved | 89 (83.2): 15 (14.0): 3 (2.8) |
| **yp Stage** | |
| 0: 1: 2: 3 | 33 (30.8): 8 (7.5): 38 (35.5): 28 (26.2) |
| **ypT category** | |
| 0: Tis: 1: 2: 3: 4 | 33 (30.8): 2 (1.9): 12 (11.2): 16 (15.0): 42 (39.3): 2 (1.9) |
| **ypN category** | |
| 0: 1: 2: 3 | 67 (62.6): 31 (29.0): 6 (5.6): 3 (2.8) |
| **First failure site** | |
| Locoregional | 23 (21.5) |
| Distant | 15 (14.0) |
| Locoregional + distant | 19 (17.8) |
| Disease free | 50 (46.7) |

## Factors associated with overall survival

The median follow-up period and OS were 18.4 (IQR, 8.1–40.7) and 24.2 months (IQR, 11.1–93.9), respectively. Median OS was inferior in patients with a baseline LN volume > 7.7 ml (18.8 vs. 46.9 months, p = 0.049; Fig 1A) and TRG 3–5 after CCRT (13.9 vs. 86.7 months, p < 0.001; Fig 1B), but longer in cases with pCR (93.9 vs. 17.6 months, p < 0.001; Fig 1C). There was no association between primary tumor volume and OS (p = 0.363). Furthermore, large LN volume (HR 1.753, 95% CI 1.015–3.029, p = 0.044) and high TRG (HR 3.276, 95% CI 1.556–6.898, p = 0.002) were confirmed as poor prognostic factors by multivariate analysis (Table 3), but pCR did not independently predict OS.

## Large LN volume associated with inferior locoregional control and survival

We conducted further investigation to understand why a large LN volume resulted in inferior OS. Regarding to the first cancer relapse, 23 (21.5%) patients experienced locoregional recurrence, and 15 (14.0%) had distant metastases (Table 2). Additionally, locoregional and distant recurrence occurred simultaneously in 19 (17.8%) cases. During follow-up, 50 (46.7%) patients remained free of cancer relapse. A LN volume > 7.7 ml was associated with higher locoregional failure (HR 1.938, 95% CI 1.044–3.599, p = 0.033; Fig 2A) and inferior DFS (HR 1.676, 95% CI 1.046–2.683, p = 0.041; Fig 2B), but it did not correlate with distant failure (p = 0.265). Furthermore, a radiation dose higher than 41.4 Gy was not associated with improved DFS (p = 0.462) or locoregional failure (p = 0.813) among cases with a large LN volume. Additionally, PET-CT was carried out in 45 (42.1%) patients before neoadjuvant CCRT. It was not associated with locoregional failure (p = 0.808), DFS (p = 0.742), and OS (p = 0.804).

## Discussion

This study analyzed 107 ESCC patients undergoing neoadjuvant CCRT and surgery. LN volume at baseline and TRG score at resection were independent prognostic factors for OS.

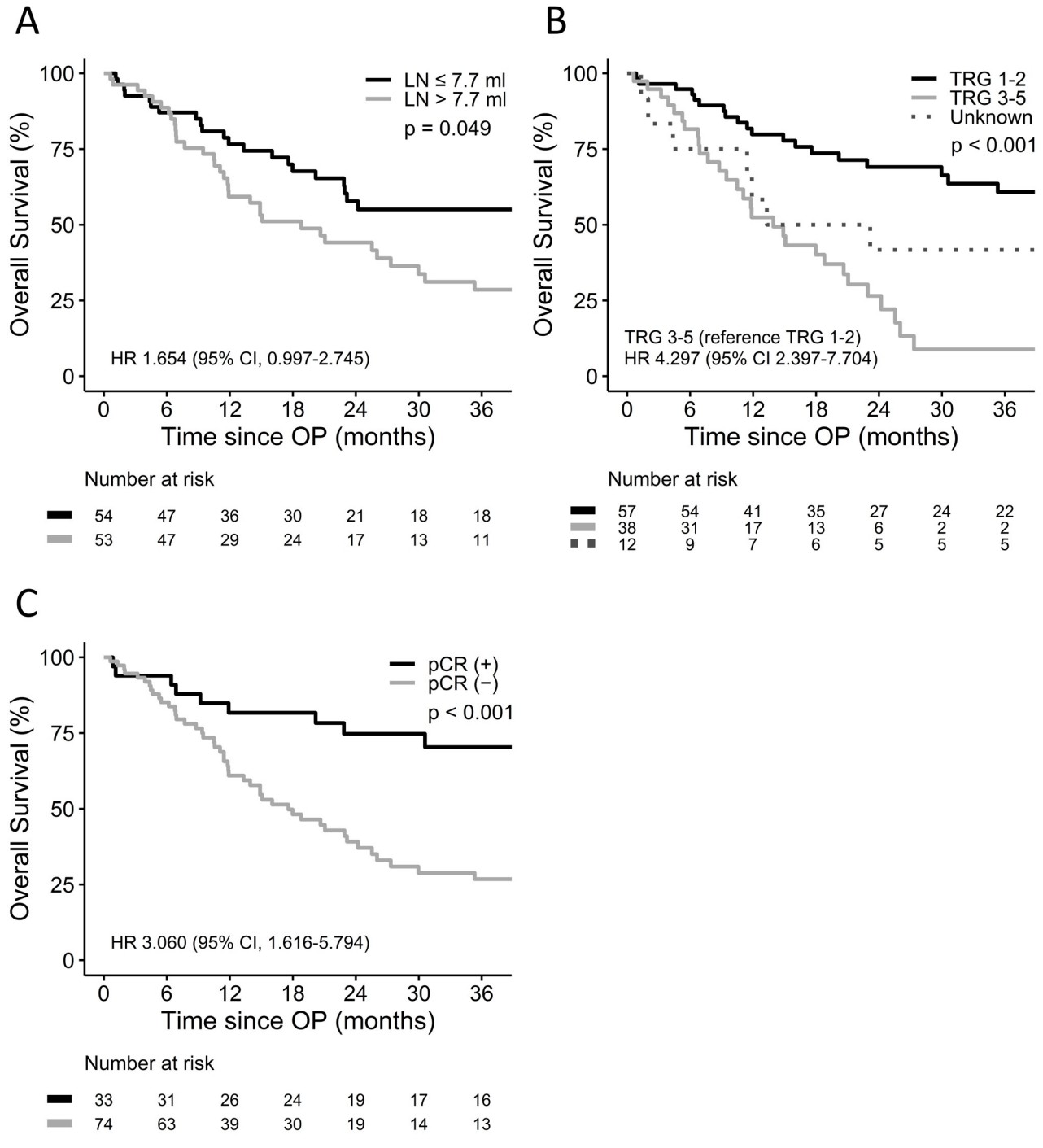

**Fig 1.** Overall survival according to (A) baseline LN volume, (B) TRG and (C) pCR status.

Importantly, there existed higher locoregional failure and inferior DFS among patients with baseline LN volume larger than 7.7 ml.

Consistent with randomized trials of esophageal cancer managed with trimodality therapy [3, 8], recurrences occurred among 53.3% of our patients during follow-up. Unlike our

**Table 3. Univariate and multivariate analyses of variables associated with overall survival.**

| Variables | Univariate | | Multivariate | |
|---|---|---|---|---|
| | HR (95% CI) | *P value* | HR (95% CI) | *P value* |
| **Age (years)** | | | | |
| ≤ 54 vs. > 54 | 1.218 (0.739–2.008) | 0.440 | | |
| **Gender** | | | | |
| Female vs. male | 0.516 (0.162–1.648) | 0.264 | 0.884 (0.265–2.956) | 0.842 |
| **ECOG PS** | | | | |
| 0–1 vs. 2 | 0.840 (0.446–1.584) | 0.590 | | |
| **cStage** | | | | |
| I&II vs. III | 0.863 (0.392–1.900) | 0.714 | | |
| **Tumor location** | | | | |
| U vs. others | 0.872 (0.472–1.609) | 0.661 | | |
| **Tumor length (cm)** | | | | |
| ≤ 5 vs. >5 | 0.772 (0.467–1.275) | 0.311 | | |
| **Esophageal tumor volume (ml)** | | | | |
| > 42.2 vs. ≤ 42.2 | 1.263 (0.762–2.094) | 0.365 | | |
| **Lymph node volume (ml)** | | | | |
| > 7.7 vs. ≤ 7.7 | 1.654 (0.997–2.745) | 0.051 | 1.753 (1.015–3.029) | 0.044 |
| **PET-CT** | | | | |
| No vs. Yes | 1.067 (0.637–1.788) | 0.804 | | |
| **Smoking** | | | | |
| No vs. Yes | 0.745 (0.321–1.729) | 0.493 | | |
| **Alcohol** | | | | |
| No vs. Yes | 0.637 (0.255–1.591) | 0.334 | | |
| **Radiation dose (Gy)** | | | | |
| ≤ 41.4 vs. > 41.4 | 0.639 (0.316–1.294) | 0.214 | 0.582 (0.279–1.211) | 0.148 |
| **Pathological complete response** | | | | |
| No vs. Yes | 3.060 (1.616–5.794) | 0.001 | 1.398 (0.591–3.307) | 0.446 |
| **Tumor regression grade** | | | | |
| 1–2 | Ref | | | |
| 3–5 | 4.297 (2.397–7.704) | 0.000 | 3.276 (1.556–6.898) | 0.002 |
| Unknown | 2.025 (0.930–4.408) | 0.076 | 1.913 (0.692–5.289) | 0.211 |
| **Surgical margin** | | | | |
| Negative | Ref | | | |
| Close | 1.436 (0.745–2.770) | 0.280 | 1.045 (0.499–2.185) | 0.908 |
| Involved | 2.660 (0.820–8.633) | 0.103 | 2.617 (0.738–9.285) | 0.136 |

ECOG PS, Eastern Cooperative Oncology Group performance status; U, upper thoracic and cervical esophagus.

previous research that showed a correlation between large primary tumor volume and inferior OS in ESCC patients receiving definitive CCRT [4], the current study did not find such an association after neoadjuvant CCRT and surgery. This result possibly reflected the high complete resection rate of the primary tumor in the current cohort. However, large baseline LN volume was related to more locoregional failure, inferior DFS, and worse OS. This finding was similar to those observed in locally advanced lung cancer treated with CCRT plus surgery [9]. To the best of our knowledge, we are the first to demonstrate that a large LN volume at baseline independently predicts survival in ESCC after trimodality therapy. The median number of LN removed at surgery for the entire cohort and the 42 patients who experienced locoregional

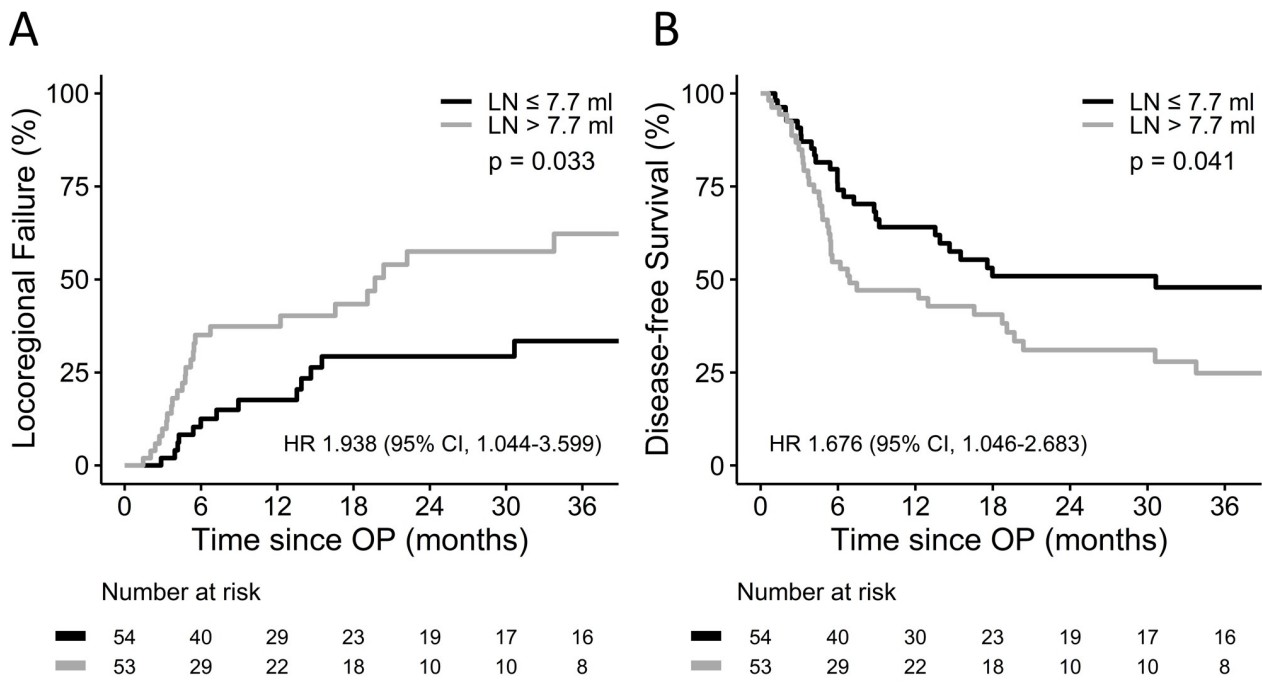

**Fig 2.** (A) Cumulative incidence of locoregional failure and (B) disease–free survival according to LN volume.

relapse as the first progression was 21.0 (IQR, 15.0–26.0) and 21.5 (IQR, 16.8–26.0), respectively. Based on the dissected LN number, lymphadenectomy appeared to be adequate in the present study [10, 11]. Although CCRT induced regression of cancer cells in the LN, it might cause fibrosis of the nodes, making residual cancer cells adhere to the surrounding tissues, and leading to locoregional relapse. We speculate that a larger LN volume implies a higher tumor burden and carries a higher probability of residual cancer cells over the dissected fields. Therefore, a large LN volume leads to higher cancer relapse and worse survival.

Esophageal cancer has the tendency to spread longitudinally within and beyond the esophageal wall. To cover the microscopic disease, several centimeters of expansion cephalad and caudal to GTV of the primary tumor for CTV are recommended. On the other hand, a margin of about 1-cm is suggested for CTV of the involved LN [12, 13]. In real-world practice, the primary gross tumor and involved LN have to be delineated separately since the margins for their CTV are different. By utilizing modern radiotherapy planning software, we can estimate the volume of the delineated involved LN. The present study demonstrated that the LN volume, a clinically available data, could be used as a prognostic factor for ESCC treated with neoadjuvant CCRT and surgery. The worse locoregional control and survival highlighted the importance of adjuvant therapies among patients with large LN volume. However, high radiation dose was not found to improve the outcomes of patients in the present study. Therefore, adjuvant immunotherapy should be particularly considered for these patients since nivolumab was associated with a significantly longer DFS than placebo for resected esophageal cancer after neoadjuvant CCRT [14].

In addition to LN volume, TRG was identified as an independent prognostic factor for OS in this study. Tumor regression in the post-therapy surgical specimens reflected individual patient sensitivity to CCRT and has been shown to correlate with survival in esophageal cancer

[15–18]. Approximately one-third of patients in our cohort achieved complete tumor regression, a result that is comparable to the 19–49% reported among ESCC patients in the literature [2, 19]. Patients who achieved pCR had significantly better survival. Notably, more than half of our patients had TRG 1 or 2 according to the Mandard classification system, which is consistent with the previous researches [17, 18, 20–22]. Furthermore, tumor down-staging was observed in two-thirds of our patients, and the down-staging rate was consistent with 50–68% shown in previous reports [20, 23, 24]. Overall, our study confirms that tumor regression after neoadjuvant CCRT is an important factor affecting the outcomes of ESCC managed with tri-modality therapy.

Our study was limited by the retrospective research design. As only ESCC patients undergoing neoadjuvant CCRT with IMRT technique were included, the results could not be generalized to patients with esophageal adenocarcinoma or those treated with definitive CCRT. But on the positive side, we provided specific information regarding clinicopathological prognostic factors and failure patterns after neoadjuvant CCRT and surgery for ESCC. Further validation in larger, independent cohorts is warranted.

## Conclusions

Large LN volume measured through radiotherapy planning CT scan predicted inferior locoregional control and survival in ESCC treated with neoadjuvant CCRT and esophagectomy.

## Supporting information

**S1 Fig. The representative images of primary tumor and LN delineation and volume measurement.** (A) The cross-sectional view of primary tumor (red color) and LN (orange color). (B)The coronal view of primary tumor (red color) and LN (orange color). (C)The 3D construction for volume measurement of primary tumor (red color) and LN (orange color). (PDF)

**S1 Dataset.**
(XLSX)

## Author Contributions

**Conceptualization:** Tzu-Hui Pao, Forn-Chia Lin.

**Data curation:** Tzu-Hui Pao, Ying-Yuan Chen, Wei-Lun Chang, Shang-Yin Wu, Wu-Wei Lai, Yau-Lin Tseng, Ta-Jung Chung.

**Formal analysis:** Tzu-Hui Pao, Forn-Chia Lin.

**Funding acquisition:** Tzu-Hui Pao, Forn-Chia Lin.

**Investigation:** Tzu-Hui Pao, Ying-Yuan Chen, Forn-Chia Lin.

**Methodology:** Tzu-Hui Pao, Forn-Chia Lin.

**Resources:** Ying-Yuan Chen, Wei-Lun Chang, Shang-Yin Wu, Wu-Wei Lai, Yau-Lin Tseng, Ta-Jung Chung.

**Supervision:** Forn-Chia Lin.

**Validation:** Ying-Yuan Chen, Wei-Lun Chang.

**Visualization:** Tzu-Hui Pao.

**Writing – original draft:** Tzu-Hui Pao, Forn-Chia Lin.

**Writing – review & editing:** Forn-Chia Lin.

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
