## [Decision Letter · Decision Letter 0]

6 Dec 2023

PONE-D-23-16460Lymph node volume predicts survival in esophageal squamous cell carcinoma treated with neoadjuvant chemoradiotherapy and surgeryPLOS ONE

Dear Dr. Lin,

Thank you for submitting your manuscript to PLOS ONE. After careful consideration, we feel that it has merit but does not fully meet PLOS ONE’s publication criteria as it currently stands. Therefore, we invite you to submit a revised version of the manuscript that addresses the points raised during the review process.

We look forward to receiving your revised manuscript.

Kind regards,

Andrea D’Aviero

Academic Editor

PLOS ONE

3. Thank you for submitting your work to PLOS ONE. Before we can proceed, please report in the Methods section the dates when data were accessed for research purposes. Please note that if this information is not included when your manuscript is resubmitted, it may be rejected

Additional Editor Comments:

The article is clearly written and provides an interesting analysis of the correlation of nodal volume and survival in SCC esophageal cancer. This should represent the basis of a prospective project.

Reviewers' comments:

Reviewer's Responses to Questions

**Comments to the Author**

1. Is the manuscript technically sound, and do the data support the conclusions?

Reviewer #1: Yes

Reviewer #2: Yes

Reviewer #3: Yes

2. Has the statistical analysis been performed appropriately and rigorously? 

Reviewer #1: Yes

Reviewer #2: N/A

Reviewer #3: Yes

3. Have the authors made all data underlying the findings in their manuscript fully available?

Reviewer #1: Yes

Reviewer #2: Yes

Reviewer #3: Yes

4. Is the manuscript presented in an intelligible fashion and written in standard English?

Reviewer #1: Yes

Reviewer #2: Yes

Reviewer #3: Yes

5. Review Comments to the Author

Reviewer #1: The manuscript is presented in an intelligible fashion and written in standard English. The data support the conclusions. It is an interesting work. It deserves publication. No changes needed to the article

Reviewer #2: .the work seems interesting to me and has a potential impact on clinical practice. Obviously, I would advise the authors to continue with the study perhaps considering a prospective study in order to eventually confirm the reported data.

Reviewer #3: Interesting work, please review this comment in order to publish.

"PET-CT was performed in cases with indeterminate results on CT or bone scan": risk of target missing should be underlined, it could be interesting to know the rate of locoregional failure among patients undergoing PET for stage assessment

6. PLOS authors have the option to publish the peer review history of their article (what does this mean?). If published, this will include your full peer review and any attached files.

Reviewer #1: No

Reviewer #2: No

Reviewer #3: No

---

## [Author Response · Author response to Decision Letter 0]

22 Jan 2024

Point-to-point Response to Editor and Reviewers 

We have made revisions in this new version according to your and the reviewers’ comments as follows:

Response to Editor (Journal requirements): 

We appreciate the valuable comments and suggestions.

Response: We ensure that our manuscript meets PLOS ONE's style requirements.

2. In your Data Availability statement, you have not specified where the minimal data set underlying the results described in your manuscript can be found. PLOS defines a study's minimal data set as the underlying data used to reach the conclusions drawn in the manuscript and any additional data required to replicate the reported study findings in their entirety. All PLOS journals require that the minimal data set be made fully available. 

Upon re-submitting your revised manuscript, please upload your study’s minimal underlying data set as either Supporting Information files or to a stable, public repository and include the relevant URLs, DOIs, or accession numbers within your revised cover letter. Any potentially identifying patient information must be fully anonymized.

Important: If there are ethical or legal restrictions to sharing your data publicly, please explain these restrictions in detail. Please see our guidelines for more information on what we consider unacceptable restrictions to publicly sharing. Note that it is not acceptable for the authors to be the sole named individuals responsible for ensuring data access.

Response: We have uploaded the minimal data set underlying the results in a supporting information file, S1 Dataset. 

3. Thank you for submitting your work to PLOS ONE. Before we can proceed, please report in the Methods section the dates when data were accessed for research purposes. Please note that if this information is not included when your manuscript is resubmitted, it may be rejected.

Response: We have reported when data were accessed for research purposes in page 5 line 79-81 of our revised manuscript.

Response: We ensure that our reference list is complete and correct. There was no change to the reference list in this revised manuscript. 

5. Additional Editor Comments: The article is clearly written and provides an interesting analysis of the correlation of nodal volume and survival in SCC esophageal cancer. This should represent the basis of a prospective project. 

Response: We appreciate the valuable comments.

Response to Reviewers:

Reviewer 1: The manuscript is presented in an intelligible fashion and written in standard English. The data support the conclusions. It is an interesting work. It deserves publication. No changes needed to the article.

Response: We appreciate the valuable comments.

Reviewer 2: The work seems interesting to me and has a potential impact on clinical practice. Obviously, I would advise the authors to continue with the study perhaps considering a prospective study in order to eventually confirm the reported data.

Response: The valuable comments and suggestions are highly appreciated. We will consider prospective studies to validate our findings. 

Reviewer 3: Interesting work, please review this comment in order to publish.

"PET-CT was performed in cases with indeterminate results on CT or bone scan": risk of target missing should be underlined, it could be interesting to know the rate of locoregional failure among patients undergoing PET for stage assessment.

Response: We appreciate the valuable comments. PET-CT was carried out in 45 (42.1%) patients before neoadjuvant CCRT. It was not associated with locoregional failure (p = 0.808), DFS (p = 0.742), and OS (p = 0.804). The paragraphs describing these results have been integrated into page 10 line 181-183, and the relevant data have also been added in Table 1 and Table 3 of our revised manuscript.

---

## [Decision Letter · Decision Letter 1]

23 Feb 2024

Lymph node volume predicts survival in esophageal squamous cell carcinoma treated with neoadjuvant chemoradiotherapy and surgery

PONE-D-23-16460R1

Dear Dr. Lin,

We’re pleased to inform you that your manuscript has been judged scientifically suitable for publication and will be formally accepted for publication once it meets all outstanding technical requirements.

Kind regards,

Andrea D’Aviero

Academic Editor

PLOS ONE

Additional Editor Comments (optional):

The authors have provided the requested revisions and is actually woth of acceptance.

Reviewers' comments:

Reviewer's Responses to Questions

**Comments to the Author**

1. If the authors have adequately addressed your comments raised in a previous round of review and you feel that this manuscript is now acceptable for publication, you may indicate that here to bypass the “Comments to the Author” section, enter your conflict of interest statement in the “Confidential to Editor” section, and submit your "Accept" recommendation.

Reviewer #1: All comments have been addressed

Reviewer #3: (No Response)

2. Is the manuscript technically sound, and do the data support the conclusions?

Reviewer #1: Yes

Reviewer #3: (No Response)

3. Has the statistical analysis been performed appropriately and rigorously? 

Reviewer #1: Yes

Reviewer #3: (No Response)

4. Have the authors made all data underlying the findings in their manuscript fully available?

Reviewer #1: Yes

Reviewer #3: (No Response)

5. Is the manuscript presented in an intelligible fashion and written in standard English?

Reviewer #1: Yes

Reviewer #3: (No Response)

6. Review Comments to the Author

Reviewer #1: The submitted work is well written and worthy of publication. The topic covered was well stated and the data consistent with the literature on the subject. The paper is worthy of publication. No revisions required

Reviewer #3: (No Response)

7. PLOS authors have the option to publish the peer review history of their article (what does this mean?). If published, this will include your full peer review and any attached files.

Reviewer #1: No

Reviewer #3: No

---

## [Editor Report · Acceptance letter]

19 Mar 2024

PONE-D-23-16460R1 

PLOS ONE

Dear Dr. Lin, 

I'm pleased to inform you that your manuscript has been deemed suitable for publication in PLOS ONE. Congratulations! Your manuscript is now being handed over to our production team.

Kind regards, 

on behalf of

Dr. Andrea D’Aviero 

Academic Editor

PLOS ONE